# Empowering Potential of the My Assistive Technology Guide: Exploring Experiences and User Perspectives

Elsa M. Orellano-Colon [1,*], Angelis Fernández-Torres [1], Nixmarie Figueroa-Alvira [1], Bernice Ortiz-Vélez [1], Nina L. Rivera-Rivera [1], Gabriela A. Torres-Ferrer [1] and Rubén Martín-Payo [2]

[1]   Occupational Therapy Program, School of Health Professions, University of Puerto Rico Medical Sciences Campus, P.O. Box 365067, San Juan 00936-5067, Puerto Rico; angelis.fernandez@upr.edu (A.F.-T.); nixmarie.figueroa@upr.edu (N.F.-A.); bernice.ortiz@upr.edu (B.O.-V.); nina.rivera1@upr.edu (N.L.R.-R.); gabriela.torres28@upr.edu (G.A.T.-F.)

[2]   Faculty of Medicine and Health Sciences, University of Oviedo, Calle de Valentín Andrés Alvarez, 33006 Oviedo, Spain; martinruben@uniovi.es

*   Correspondence: elsa.orellano@upr.edu; Tel.: +1-(787)-758-2525 (ext. 4200)

**Abstract:** The use of assistive technology (AT) devices enhances older adults' function in daily activities. However, the lack of awareness of AT among potential AT users has been identified as a major barrier to its adoption. This study aimed to assess the quality of the Mi Guía de Asistencia Tecnológica (MGAT) web app, which provides information on AT, from the perspective of older Latinos with physical disabilities, and to explore their experience using the MGAT. We employed a convergent parallel mixed-method design involving 12 older Latinos living in Puerto Rico. In Phase I, the researchers trained the participants in the use of the MGAT. In Phase II, participants were encouraged to use it for 30 days. In Phase III, the participants completed the User Mobile Application Rating Scale (uMARS) and individual interviews, analyzed with descriptive statistics and a directed thematic content analysis. The quality of the MGAT was rated high in both the objective (uMARS mean = 3.99; SD = 0.68) and subjective (uMARS mean = 4.13; SD = 1.10) domains. Qualitative data revealed the MGAT was accessible, usable, desirable, credible, useful, and valuable to increasing older Latinos' AT knowledge, function, and autonomy. The MGAT has the potential to increase AT awareness and adoption among older adults.

**Keywords:** activities of daily living; assistive technology; Latinos; older adults; web app

## 1. Introduction

Functional disabilities (FDs) are defined as any difficulty in performing or participating in daily life activities essential for independent living, such as walking, bathing, or completing household tasks [1]. Physical FDs in older adults can arise from the normal aging process or age-related chronic conditions like arthritis, diabetes, hypertension, stroke, heart diseases, or pulmonary diseases [2]. FDs pose a significant public health challenge, disproportionately affecting older adults in the continental United States (U.S.) and Puerto Rico (P.R.). National data reveal a notably higher percentage of older adults (65 years and older) with disabilities in daily activities among Latinos in P.R. (26.7%) compared to their counterparts in the U.S. (13.2%) [3].

This increased prevalence of FDs among older adults in P.R. heightens their vulnerability to a decreased ability to perform and participate in activities of daily living (ADLs) and instrumental activities of daily living (IADLs), leading to increased dependence, reduced quality of life, poor self-efficacy, fear of falling, falls and injury, social isolation, depression, and diminished health levels [4,5]. Contextual factors like stigmatization, discrimination, and societal inequality can also contribute to functional disabilities [6]. Economic factors, inadequate nutrition and healthcare, unsuitable housing settings, and limited access to employment and education in the country can result in disabilities, significantly impacting

quality of life for individuals with functional disabilities. Furthermore, the absence of support systems and cultural values are contextual factors playing a crucial role in people's occupational performance and participation, contributing to disabilities [7].

At the population level, a free assistive technology (AT) guide app may contribute to addressing the predominant barrier to using AT devices among older Latinos with physical function disabilities living in P.R.: the lack of access to knowledge about AT devices. This is crucial, as research has shown that information about how to acquire AT devices is the most important prerequisite for the use of AT [8].

The role of function as a crucial determinant of health [9] is underscored by its impact on the well-being of older adults. The utilization of low-tech AT devices, such as jar openers, shower benches, and mobility aids, proves instrumental in enhancing functional independence. By mitigating physical limitations, these devices empower older individuals to engage in meaningful activities and occupations [10–13]. Furthermore, the adoption of AT devices holds the potential to decrease the likelihood of institutionalization, reduce healthcare service costs, and elevate safety, thereby contributing to the overall well-being and quality of life for this demographic [12]. Conversely, reluctance or non-use of AT devices among older adults may lead to compromised health and limitations in community participation [14]. This underscores the importance of embracing assistive technologies to optimize health outcomes and overall quality of life for this population.

Despite the benefits of using AT devices, studies indicate that Latinos exhibit the lowest likelihood of utilizing and obtaining AT devices [15]. A previous systematic review revealed that AT users lacked reliable information and awareness about assistive technology, which is crucial in the context of behavior change for successful AT adoption and use [16]. In Puerto Rico, our previous investigations with older Latinos revealed that a deficiency in access to information about AT devices and services was the predominant barrier to acquiring and using AT, surpassing all other reported barriers [17,18].

The combination of a high prevalence of FDs and the lack of access to AT information poses significant challenges to the provision of equitable access to effective AT services and support. A partial solution involves digital AT resources. The potential of mobile health (mHealth) technologies to enhance healthcare accessibility, particularly for resource-constrained demographics like older adults, is substantial in the digital era [19]. However, despite the availability of numerous health applications designed to assist older individuals [20], none were identified in the Android Google Play store or iPhone Apple App store specifically tailored to provide information about assistive technology devices for older adults with physical function disabilities in activities of daily living (ADLs) and instrumental activities of daily living (IADLs).

To address this gap, in 2022, the Occupational Therapy Program of the University of Puerto Rico Medical Sciences Campus empirically developed a free informational web application, "Mi Guía de Asistencia Tecnológica" (MGAT; My Assistive Technology Guide), targeted at older Latinos with physical function limitations to provide detailed information about assistive technology devices for activities of daily living (ADLs) and instrumental activities of daily living (IADLs) [21]. While the MGAT serves as a valuable resource for information, it does not function as a decision-making tool. In other words, the MGAT does not provide guidance to users on selecting the appropriate assistive technology tailored to their individual needs. Our previous research found that the MGAT prototype was a usable and acceptable resource for providing information about AT devices and services [21]. Still, the multidimensional quality of the MGAT from the perspective of older adult end-users has not been established. User-centered app evaluations are crucial to improve app design, effectiveness, and, importantly, app adherence [22].

One of the most comprehensive and multidimensional tools to evaluate app quality is the User Mobile App Rating Scale (uMARS) [23]. This scale provides comprehensive ratings of user experience and impressions of the app by assessing app quality (objective and subjective) and perceived impact. It consists of twenty items, organized in five subscales, to evaluate the quality of health apps. Four subscales are objective and related to quality

rating—engagement (five items), functionality (four items), aesthetics (three items), and information (four items)—and one subscale evaluates app quality subjectively (four items). A further subscale of six items measures the user-perceived impact of the evaluated app. Each item has customized wording appropriate to the aspect being assessed. Items employ a common 5-point rating scale from 1 (Inadequate) to 5 (Excellent), such that higher scores represent a stronger impact of the app on that aspect of user cognition and/or potential behavior.

The uMARS has been used in several older adults/end users' reviews of apps for a variety of behaviors, such as the self-management of heart failure in palliative care [24], exercise rehabilitation using wearable sensor-based biofeedback [25], gout self-management [26], and post-traumatic stress disorder self-management [27]. Therefore, the aims of the current study were to assess the quality of the MGAT by older Latinos with physical function disabilities using uMARS and to explore how this population thinks, feels, and uses the MGAT to access information about AT devices and services for ADLs and IADLs.

## 2. Materials and Methods

We applied a convergent parallel mixed-method design to gain in-depth understanding of the older adults' experiences with the MGAT [28]. This design included the simultaneous collection of both qualitative (QUAL) and quantitative (QUAN) data in one phase of the study, weighing each method equally, analyzing each datum separately, and then merging both sources of data to interpret the findings. The study was approved by the Institutional Review Board of the University of Puerto Rico, Medical Sciences Campus, and conducted referring to good clinical practice. All participants provided informed consent and were asked to use the MGAT for a period of 30 days. At the end of the trial, we collected quantitative data using the User Mobile App Rating Scale (uMARS), with their items categorized by the dimensions of the honeycomb model, and qualitative data using a semi-structured interview to assess the MGAT's utility, usefulness, and acceptability with open-ended questions that explain their answers to the uMARS items.

### 2.1. Karagianni's Optimized Honeycomb Model

Karagianni's Optimized Honeycomb model, originally developed by Morville, was used to guide this study's data collection and analysis [29,30]. This model is employed to elucidate the user's experience and perspective when designing applications. It describes seven facets grouped into three categories, reasoning about how the users "Feel, Think, and Use" the application by making a connection between the seven facets. Karagianni's Optimized Honeycomb model provided an adequate structure to collect and analyze data of the MGAT's utility, usefulness, and acceptability among older adult users. See Table 1 for the definitions of the seven facets grouped into the three categories of the model and its application to this study.

**Table 1.** Definitions of the seven faucets of Karagianni's Optimized Honeycomb model.

| Categories | Facets | Descriptions and Definitions |
|---|---|---|
| Use | Findable | Navigable and easy to find information. |
| | Accessible | Manageable for people with disabilities or reduced function. |
| | Usable | Ability to be used. |
| Feel | Desirable | Features with an emotional engagement. |
| | Credible | Influence of elements for users' trust and belief in the product. |
| Think | Useful | To make innovative solutions that are beneficial. |
| | Valuable | Deliver value or advance the mission. |

### 2.2. Intervention

The MGAT was designed as a traditional, easy-to-use HTML web interface with customized culturally sensitive content of assistive technology devices that could compensate for older Latinos' physical FDs in ADLs and IADLs [21]. The MGAT (Figure 1) was de-

signed with the purpose of being an informational web app to provide this population with access to and insights into various types of assistive technologies. Specifically, the MGAT allows for the user to effectively navigate through eight different areas of activities of daily living to finally learn about 97 available AT devices used to compensate for the functional disabilities that older adults may have in a variety of daily living activities. The user can explore AT devices within eight broad areas of activities of daily living in which older adults may exhibit functional difficulties using buttons with photos and text to represent each area (mobility, self-care, bathing, dressing, meal preparation, home management, medication management, and home safety). Each of these areas of activities contain sub-categories of activities pertaining to each activity. The MGAT includes the description, approximate cost, benefits, considerations, and resources for acquiring each AT device. The MGAT also includes instructional videos of older people using these devices. As an informational web app, the MGAT does not have interactive features. The MGAT is compatible with any mobile device, as an attempt to increase its free access to the end user. Although the MGAT provides valuable information of ATDs, it does not offer decision-making support for selecting the most suitable device based on individual needs. Therefore, all videos include a disclaimer stating the following: "The equipment shown in this video is only demonstrative. This video is solely for educational purposes. We recommend that you consult with your occupational therapist for an evaluation of your need for assistive technology equipment". Even though it is an ongoing project, the link to the functional prototype is available at https://www.figma.com/proto/UcJ2xECanIxELPYr9pMxPq/RCM---Design-&-Prototype?page-id=10:17&node-id=220-1527&viewport=296,67,0.08&scaling=scale-down&starting-point-node-id=220:1527 (accessed on 11 December 2023).

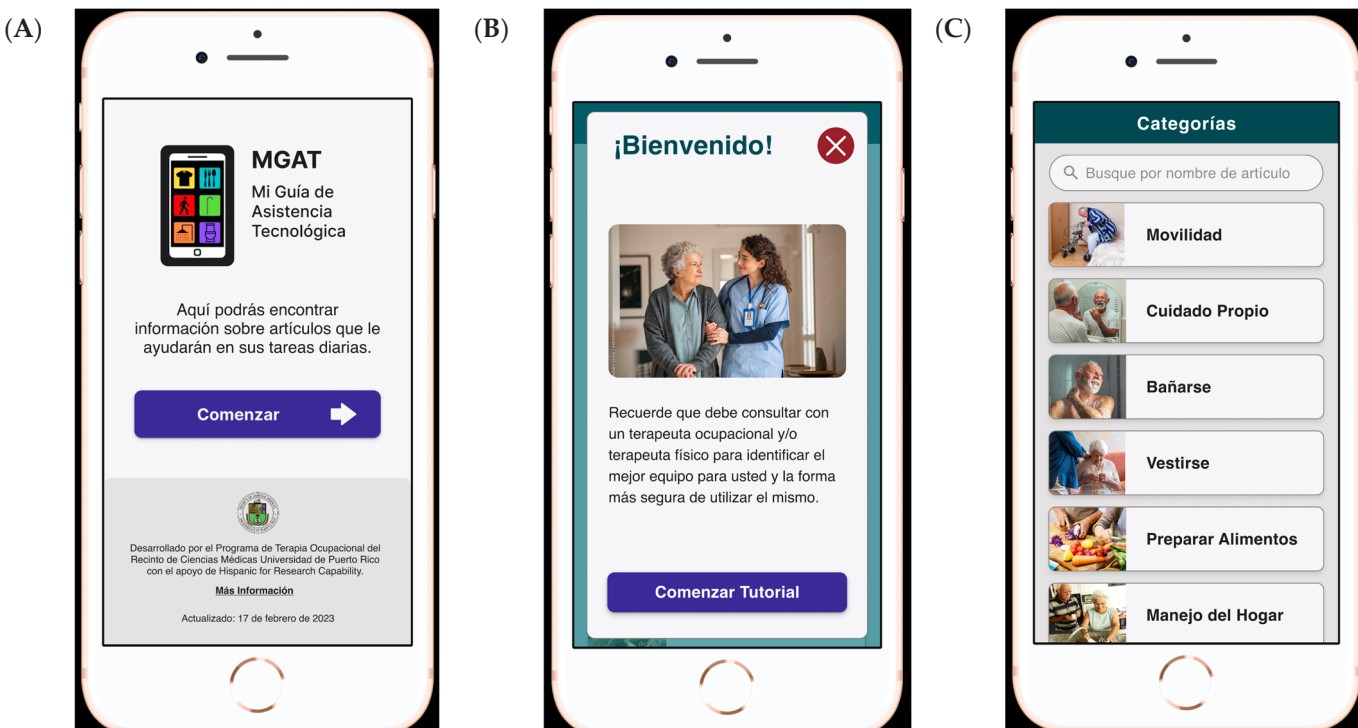

**Figure 1.** *Cont.*

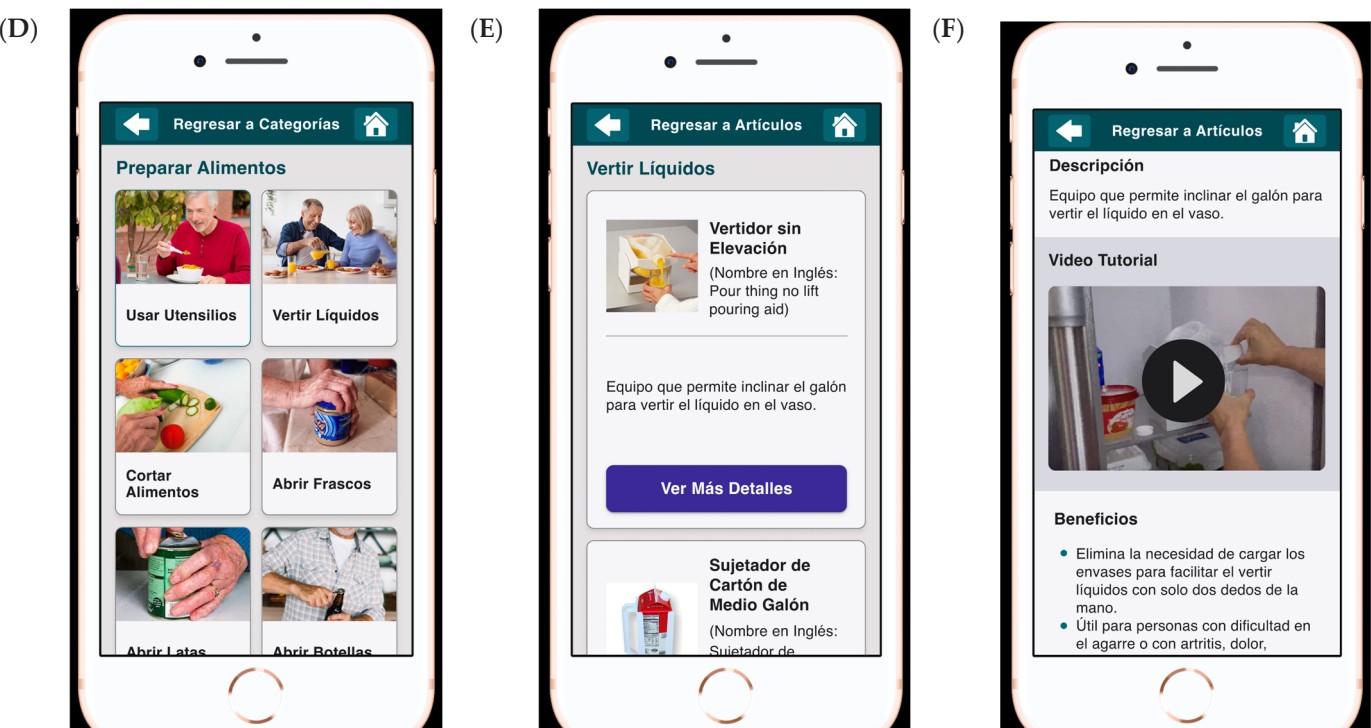

**Figure 1.** Features of the MGAT web app. (**A**) Introductory screen to the home page; (**B**) tutorial screen; (**C**) screen of categories of daily activities; (**D**) screen of activities within the selected categories of activities; (**E**) screen of assistive technology devices; and (**F**) screen of information and instructional video of the assistive technology. *Notes*: The MGAT uses Google images depicting individuals engaging in various daily activities. As the MGAT is a prototype developed for non-profit research purposes, we have employed these Google Images under the Fair Use doctrine. This doctrine permits the use of copyrighted materials without seeking permission from the copyright owner for non-profit research endeavors.

### 2.3. Participants and Recruitment

The inclusion criteria for participants were as follows: (1) being a community-living Latino adult aged 65 years and older; (2) not requiring supervision to perform their daily activities; (3) reporting difficulties in one or more activities of daily living (ADLs) or instrumental activities of daily living (IADLs); (4) having functional comprehension and verbal communication skills, evidenced by a correct understanding of this study's purpose and informed consent; (5) ownership of a smartphone, tablet, or computer with internet access; and (6) able to read in Spanish. Participants were excluded if they were receiving home healthcare services, were institutionalized, or were bedridden.

We recruited a purposive sample of 14 older adults based on Hwang and Salvendy's recommendations, suggesting that when conducting user testing, it is sufficient to use the rule of thumb of 10 ± 2 to estimate the required number of users [31]. Two participants dropped out because they did not use the MGAT during the trial period.

Participants were recruited through direct contact and snowball sampling by five occupational therapy master's students. Direct contact (in person or by phone) was arranged with participants known to the researchers who could potentially meet the study's inclusion criteria. In the snowball sampling procedure, the researchers asked participants who agreed to participate in the study to provide the researchers' telephone numbers to other potential participants who might fit the research criteria and be willing to participate. Interested individuals were then asked to contact the researchers to determine their eligibility and set up an appointment to complete the enrollment process.

*2.4. Data Collection Procedures*

Data collection took placed through face-to-face individual meetings in a private space at the participants' homes, facilitated by one of the five occupational therapy graduate student data collectors. These students underwent a two-day training in data collection procedures conducted by the principal investigator, an established researcher in aging, assistive technology, and quantitative, qualitative, and mixed-methods data collection and analysis. The data collection procedures were divided into three phases: (1) participants' MGAT training; (2) MGAT trial period; and (3) quality testing (see Figure 2).

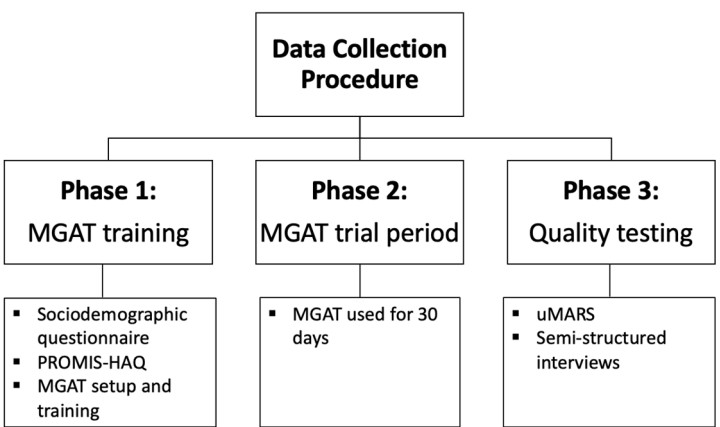

**Figure 2.** Description of the three phases of this study.

2.4.1. Phase 1: MGAT Training

An initial face-to-face meeting was organized with all consenting participants to administer the sociodemographic questionnaire and the PROMIS HAQ, aiming to describe the participants' characteristics and level of physical function [32]. Subsequently, the researchers added the MGAT to the home screen on the participants' mobile devices and provided app training for an average of 30 min. During this training, the researchers demonstrated all the features of the MGAT and guided the participants in navigating through its functionalities until they felt comfortable using it. The participants were instructed to use the app for 30 days and were advised to contact the researchers if they needed any assistance throughout this period.

2.4.2. Phase 2: MGAT Trial Period

During this phase, the participants used the MGAT app as desired for 30 days. A 30-day trial period offers older adults a practical and manageable timeframe to evaluate the utility and usefulness of the web app, accommodating their need for increased time to learn and adapt to technology. This study did not prescribe a specific frequency for app usage; instead, participants were free to determine how often they used the MGAT based on their own criteria and willingness.

2.4.3. Phase 3: Quality Testing

At the end of the trial period, the researchers scheduled a face-to-face individual meeting with the participants at their homes. During this meeting, the researchers first administered the uMARS using an interview format and simultaneously conducted individual interviews to elaborate on the participants' responses to the uMARS items. The individual interviews also included additional questions to explore the usability, usefulness, and acceptability of the MGAT. Participants provided consent for audio recording of the interviews, allowing the researchers to obtain verbatim transcripts of their expressions and gather a rich description of their opinions.

*2.5. Study Measures*

2.5.1. Sociodemographic Questionnaire

A sociodemographic questionnaire was developed by the researchers of this study to gather a sociodemographic profile of the older adult participants. It collected de-identified data related to the participant's age, sex, education level, medical conditions, marital status, city of residence, number of persons living with the participant, working status, monthly income, sources of income, healthcare plan, and use of AT devices.

2.5.2. PROMIS Physical Function 24a (PROMIS-HAQ)

The PROMIS-HAQ is a patient-reported outcome measure designed to evaluate physical function status in adults across categories such as dressing and grooming, arising, eating, walking, hygiene, reaching, gripping, errands, and chores [32]. Numerous validation studies have been conducted for the parent Health Assessment Questionnaire (HAQ), and these apply to the PROMIS-HAQ [33]. The PROMIS-HAQ items were translated into Spanish using a universal approach for translations and cultural adaptation of instruments. The PROMIS-HAQ was utilized to describe the level of physical function in the sample of this study.

2.5.3. User Version of the Mobile Application Rating Scale (uMARS) Spanish Version

For this study, the researchers utilized the Spanish version of the uMARS, which was developed by considering item and conceptual equivalences assessed by two health professionals with expertise in mHealth [34]. The Spanish uMARS has shown excellent internal consistency ($\alpha$ 0.89 and 0.67 for objective and subjective quality, respectively) and temporal stability (r > 0.82 for all items and subscales). The Spanish version of the uMARS is available from the author without a license, fee, or royalty.

2.5.4. Semi-Structured Interview Guide

The semi-structured interview guide for this study was developed by the researchers, drawing from the facets of Karagianni's Optimized Honeycomb model [29]. The guide aimed to further explore participants' responses to the uMARS items and capture their experiences, providing insights into how they felt, thought, and used the MGAT. The guide contained the following question to be asked after the participant rated each of the uMARS items: "Why did you rate this item with that score?" The guide also included the final open-ended questions: "What do you think about the MGAT? What did you like and dislike about the MGAT? What will encourage you to continue using the MGAT? What value does the MGAT have for you? What recommendations do you have for improving the MGAT?" The interviews ranged in duration from 45 to 80 min.

*2.6. Data Analysis*

Data analysis was conducted in Spanish by the researchers of this study. Quantitative data obtained from the sociodemographic questionnaire, PROMIS-HAQ, and uMARS were analyzed using central tendency descriptive statistics, presenting means and standard deviations for continuous variables and frequencies with percentages for categorical variables. Microsoft Excel 2019 facilitated these statistical analyses. For qualitative data from the semi-structured interviews, a directed thematic content analysis was employed [35].

The directed qualitative content analysis used a deductive approach guided by the Optimized Honeycomb Model by Karagianni [29]. The analysis began with operational definitions of the seven facets (useful, valuable, credible, desirable, findable, usable, and accessible) of the Optimized Honeycomb Model. The researchers independently immersed themselves in the data, highlighting text corresponding to each facet, and coding it accordingly. The predetermined categories were applied wherever possible, and any text not fitting into these categories was coded with a label summarizing the MGAT's utility. The research team engaged in regular meetings to interpret the findings collaboratively.

QDA Miner Lite V3.0.2 (a qualitative data analysis software package) facilitated data management [36].

Rigor was ensured through rich and detailed descriptions, peer debriefing sessions with two occupational therapy student researchers per participant, and an external audit by a qualitative data analysis expert (the PI) [37]. This external audit involved reviewing all interviews to validate interpretations and data analysis conducted by the occupational therapy student researchers.

Following the completion of both qualitative and quantitative analyses, the qualitative findings from individual interviews were integrated with the quantitative results of the uMARS for comprehensive analysis and comparisons. A narrative integration approach was applied, weaving qualitative and quantitative findings together on a theme-by-theme or concept-by-concept basis to offer a more holistic understanding of participants' experiences [38]. The quantitative and qualitative data analysis was conducted in Spanish, and the specific quotes selected to illustrate each theme were translated into English by a bilingual researcher for inclusion in the manuscript. This approach ensures that the original meaning and context of the quotes are preserved in the translation, while also making the findings accessible to an English-speaking audience.

## 3. Results

### 3.1. Participants' Characteristics

Fifteen older adults were initially assessed for eligibility, with fourteen found to be eligible for participation. One participant was excluded due to an inability to correctly articulate the purpose and procedures of the study. Additionally, two participants dropped out because they forgot to use the MGAT during the 30-day trial period. None of the participants contacted the researchers for assistance in using the MGAT throughout the trial.

The individual characteristics of the 12 participants who completed the study, including T-scores and the severity of physical function disability, are detailed in Table 2. Seven (58%) participants were women. The average age of the participants was 74 years, ranging from 65 to 86 years. All participants reported an education level of a high school degree or higher (100%). The most frequently reported physical conditions included hypertension ($n = 9$, 75%), diabetes ($n = 5$, 41.7%), arthritis ($n = 4$, 33.3%), and gastrointestinal conditions ($n = 4$, 33.3%). The majority of participants were married ($n = 6$, 50%), and most had Medicare healthcare plans ($n = 10$, 83.3%). Two participants reported an annual income of less than USD 1000 (16.7%), six participants reported an annual income of USD 1000 to USD 2000 (50%), and four participants reported an annual income of more than USD 2000 (33.3%).

**Table 2.** A snapshot of participants self-reported physical function and characteristics.

| Participant's Number | PROMIS HAQ (T-Score; Severity) [a] | Sex | Age | Health Conditions | Living Situation | Highest Level of Education | Annual Income |
|---|---|---|---|---|---|---|---|
| A2 | 37.6; moderate | M | 86 | Diabetes Hypertension Heart disease | Married | High school graduate | USD 1700 |
| A3 | 58.1; above average | F | 70 | Hypertension Visual | Divorced; separated | Bachelor's degree | USD 1800 |
| B1 | 58.1; above average | M | 73 | Diabetes Hypertension | Widowed | Bachelor's degree | USD 3000 |
| B2 | 58.1; above average | M | 74 | Other | Married | Bachelor's degree | USD 6000 |
| B3 | 43.4; mild | F | 65 | Obesity Hypertension Gastrointestinal disease Respiratory disease Arthritis | Divorced; separated | High school graduate | USD 600 |

**Table 2.** *Cont.*

| Participant's Number | PROMIS HAQ (T-Score; Severity) [a] | Sex | Age | Health Conditions | Living Situation | Highest Level of Education | Annual Income |
|---|---|---|---|---|---|---|---|
| G1 | 46.7; above average | M | 77 | Diabetes Hypertension Obesity Heart disease Gastrointestinal disease Arthritis Depression Other | Married | High school graduate | USD 2012 |
| G2 | 48.4; above average | F | 73 | Diabetes Hypertension Obesity Gastrointestinal disease Other | Married | More than high school | USD 860 |
| L1 | 43.4; mild | F | 66 | Other | Married | Bachelor's degree | USD 2200 |
| L2 | 58.1; Above average | F | 77 | Osteoporosis Arthritis Other | Divorced; separated | High school graduate | USD 1000 |
| N1 | 58.1; above average | F | 75 | Hypertension Other | Married | More than high school | USD 2000 |
| N2 | 43.4; mild | F | 71 | Hypertension Auditory | Widowed | More than high school | USD 1887 |
| N3 | 58.1; above average | M | 81 | Diabetes Hypertension Gastrointestinal disease Arthritis | Divorced; separated | Bachelor's degree | USD 1500 |

[a] The PROMIS Health Assessment Questionnaire Disability Index uses a T-score metric in which 50 is the mean of a relevant reference population and 10 is the standard deviation (SD) of that population. Higher scores equal more of the concept being measured. We used the following PROMIS score cutoff points: mild = 45–40; moderate = 39–30; severe < 30.

### 3.2. Quantitative Results

The PROMIS-HAQ T-scores ranged from 37.6 (1.2 SD below the general population in the U.S., indicating moderate levels of impairment in physical function) to 58.1 (0.8 SD above the general population in the U.S., indicating above-average levels of physical function), with an average T-score of 50.96 (indicating an average level of physical function).

The results of the app quality evaluation are presented in Table 3. The overall uMARS objective mean score was high (M = 3.99, SD = 0.68). The "Functionality" dimension achieved the highest mean score (M = 4.40, SD = 0.89), while the "Engagement" dimension obtained the lowest mean score (M = 2.97, SD = 1.66). The overall subjective quality of the app, encompassing the subjective quality and perceived impact, was also rated high (M = 4.13, SD = 1.10).

**Table 3.** Results from the user version of the Mobile Application Rating Scale (uMARS). Overall uMARS quality score shown in bold.

| uMARS Sections | uMARS Dimensions | Mean (SD) | Range [a] |
|---|---|---|---|
| Objective | Engagement | 2.97 (1.66) | 1–5 |
| | Functionality | 4.40 (0.89) | 3–5 |
| | Aesthetics | 4.28 (1.61) | 3–5 |
| | Information | 4.31 (0.72) | 3–5 |
| | **Overall mean** | **3.99 (0.68)** | **1–5** |
| Subjective | Subjective quality | 3.56 (1.40) | 1–5 |
| | Perceived impact | 4.70 (0.81) | 3–5 |
| | **Overall mean** | **4.13 (1.10)** | **1–5** |

[a] Higher scores equal more of the concept being measured.

### 3.3. Quantitative uMARS Items Scores and Qualitative Reflections

The merging of the quantitative ratings of each uMARS item with the participants' answers to the interview questions resulted in seven themes that emerged from the Optimized Honeycomb Model and 20 sub-themes across these themes (Table 4). The sub-themes were derived from the uMARS items or domains, and the qualitative quotes emerged from the participants' explanations of their answers to the uMARS items and from the final interview questions. The average usage of the MGAT was 11.3 times during the 30-day period, with a range of 1 to 20 times.

**Table 4.** Emergent themes and sub-themes from qualitative data analysis of individual interviews with older adults participants (*n* = 12) guided by the Optimized Honeycomb Model.

| Themes | Sub-Themes |
|---|---|
| Use | |
| Findable | • Navigation<br>• Consistent gestural design |
| Accessible | • Visual layout |
| Usable | • Ease of use<br>• Performance |
| Feel | |
| Desirable | • Customization<br>• Visual appeal<br>• Appropriate for older adults<br>• Interactivity<br>• Entertainment<br>• Interesting<br>• Visual Graphics |
| Credible | • Quality of Information<br>• Trustworthiness<br>• Amount and conciseness of information<br>• Quality of visual information |
| Think | |
| Useful | • Useful to others<br>• Useful to self |
| Valuable | • Worth paying<br>• Empowered Independence |

#### 3.3.1. Findable

The Findable theme relates to the information in the application being easy to localize and includes the sub-themes of "navigation" and "consistent gestural design".

Navigation (uMARS item 8; M = 4.33, SD = 0.65). All the participants reported that moving between screens was easy, precise, and logical, confirmed by a moderately high uMARS item 8 mean score.

Consistent gestural design (uMARS item 9; M = 4.50, SD = 0.67). Most of the participants thought that the MGAT taps and scrolls were very consistent: "I give it a 5 (out of 5) because you can move around and it's like the same for everything" (B1). This was confirmed by a high uMARS item 9 mean score.

#### 3.3.2. Accessible

Accessible is defined as the appropriateness of the arrangement and size of buttons, icons, menus, and content on the MGAT screen and was represented by the "layout" sub-theme.

Layout (uMARS item 10; M = 4.33, SD = 0.89). Most of the participants agreed that the MAGT elements were clear and organized: "You open the screen and it clearly presents

what it is, how you're going to use it, the equipment and everything… everything is very clear to understand" (G2). This was confirmed by a moderately high uMARS item 10 mean score. Yet, to enhance the visual accessibility of the MGAT, some participants recommended to the increase the font size of the descriptions of the AT devices and enhance the visibility of the back arrow.

### 3.3.3. Usable

Usable was defined as the feeling of being able to use the MGAT and it encompasses two sub-themes: "ease of use" and "performance".

Ease of use (uMARS item 7; M = 4.50, SD = 0.90). Ease of use refers to how easy it is to learn to use the app and how clear the photos, menu, buttons, and instructions are. The participants agreed that the MGAT was simple and easy to use: "As long as, you know, the person has the capacity to see the application. Assuming the person is not disoriented, you can learn". This was confirmed by a high uMARS item 7 mean score. Only one participant who was not digitally oriented experienced some challenges understanding how to use the web app:

> *I get lost (using the MGAT), because when I try to go back, it gets all messed up, and what I do is close the screen and enter again because it confuses me. So I give it a 2 (out of 5) because the truth is, you get lost. Oh dear, I don't understand this device much, but at least it led me to read the information.*

Performance (uMARS item 6; M = 4.25, SD = 1.29). This sub-theme refers to how accurate and fast the MGAT functions and components work. In general, most of the participants stated that the MGAT was relatively fast and precise, but the speed of the MGAT functions and components was dependent on the speed of the participants' Wi-Fi connection: "They are fast (the buttons and menus) but sometimes the internet at home doesn't work and it becomes slow. But I think it's the signal here". This was confirmed and expanded by a moderately high uMARS item 6 mean score.

### 3.3.4. Desirable

The Desirable theme refers to how the information provided in the application is attractive and interesting to the user that navigates the platform, linking directly to an emotional engagement with the MGAT. Seven sub-themes emerged from desirable and are described below.

Appropriate for older adults (uMARS item 5; M = 4.25, SD = 1.06). This sub-theme refers to whether the photos, the videos, and the content about AT devices are suitable for people like them. The vast majority of the participants stated that the MGAT was designed specifically for older people like them with functional needs, which was confirmed by a moderately high uMARS item 5 mean score: "The people in the videos are like me and demonstrate how to use the equipment" (B3). However, one younger participant (L1; 66 years) and another with an above average level of function (B2) stated that the MGAT was mainly inappropriate for them because "Those are equipment for older people; I don't think it applies to me yet" (B2).

Customization (uMARS item 5; M = 1.25, SD = 0.62). This sub-theme refers to how the MGAT allows for the older adult to make changes to its settings and preferences such as the sound, the photos, or the content of the MGAT. There was unanimous agreement that the application does not provide the opportunity to customize any of the MGAT settings: "I didn't see that it had anything for making any changes" (N2), which was confirmed by a low mean score in this uMARS item.

Entertainment (uMARS item 1; M = 3.83, SD = 1.11). This sub-theme refers to whether the application is fun or entertaining to use. Although not vigorously found to be fun (evidenced by a moderate uMARS item 1 mean score), its educational potential was acknowledged: "It's entertaining because it's interesting to see things that one is not accustomed to seeing all the time and things that are useful, which I didn't know" (N2). One participant believed that the MGAT was more an educational rather than an entertainment tool: "I

can't say it's entertaining because it doesn't entertain me; it's more of a means to search for useful information" (A3). For some of the participants, the AT devices and videos were also found entertaining: "One gets wrapped up in looking at the little things they have there (AT devices) and watching the little videos" (B1).

Interactivity (uMARS item 4; M = 1.33, SD = 0.78). This sub-theme pertains to whether the MGAT allows for users to input information, provides feedback, or includes reminders or notifications. All of the participants agreed that the MGAT did not has interactive features: "It has no interaction with me, doesn't say anything. It also doesn't send notifications to the phone, that's why one forgets to use it" (B2). This was confirmed by a low mean score in this uMARS item.

Interesting (uMARS item 4; M = 4.17, SD = 1.03). Interesting refers to whether the application is interesting to use. Most of the participants indicated that the MGAT was interesting because they could learn about AT devices that could support their performance in daily activities: "(The MGAT) is quite interesting because it has various devices and gives you ideas on how to use them, and that's good because one uses it to do the household chores" (B1). This was confirmed by a moderately high uMARS item 4 mean score.

Visual Appeal (uMARS item 12; M = 4.42, SD = 1.24). This sub-theme refers to how well the MGAT looks. The majority of the participants found the MGAT colors, icons, pictures, and text to have an attractive appearance: "Yes, it's very nice. It doesn't look cluttered with things, and I like that" (B3). This was confirmed by a high uMARS item 12 mean score. In contrast, two participants' low ratings in this item were expanded by the following: "It looks old-fashioned, not something modern. The design and colors are poor" (L1).

Graphics (uMARS item 11; M = 4.08, SD = 1.38). This sub-theme captures the idea of how high the quality of the MGAT graphic is, including buttons, icons, photos, videos, and content. Most of the participants indicated that the visual graphics of the MGAT were professional, simple, clear, and organized: "It looks nice. The colors look good, and the buttons are large and clear" (B1). This was confirmed by a moderately high uMARS item 11 mean score. However, there were contrasting views concerning the MGAT videos' quality. Some participants rated it high: "The video looks perfect, showing how things are truly done" (N2), while two other participants rated it low due to the videos' lack of clarity and audio feedback describing the use of the AT devices.

> *(The graphics' quality) is acceptable, but its neither complete nor concise because I understand that in some explanations, more information was needed for one to be aware of what it is about. Because the videos weren't clear, and they also didn't have audio. (L2)*

3.3.5. Credible

The credible category refers to the confidence in the application, assessing whether it feels safe to use and is trustworthy. It encompasses the five sub-themes described below.

Quality of the information (uMARS item 13; M = 4.33, SD = 0.65). This sub-theme refers to whether the content of the MGAT is accurate, well written, and aligns with the objectives of the app. The participants agreed that the MGAT content was "Very good and relevant because it explains things to the point" (B5). This was confirmed by a moderately high uMARS item 13 mean score.

Quantity of information (uMARS item 14; M = 3.92, SD = 0.79). This sub-theme refers to whether the information included in the MGAT is comprehensive but concise. Most participants indicated that the information in the MGAT was comprehensive, confirmed by a moderate uMARS item 14 mean score, but recommended including links to each AT device that would guide them to purchase the item online:

> *(The MGAT) tells you everything summarized and where to buy the equipment, and I like that. . . it offers a lot of information, but it doesn't have links to more information and resources. . . You should add that. That you click on the link and it takes you to Amazon or Ebay to buy the equipment. (B1)*

Other participants recommended including additional information about strategies other than AT devices to perform daily living activities:

*I would give it a 4 (out of 5). Because there are things I would like to do without equipment, like putting on underwear that I sometimes can't reach down there. . . adding techniques on how to do these things, maybe they could also include it in the video tutorial. (G2)*

In contrast, two participants expressed divergence, feeling that the MGAT's information was insufficient and calling for the inclusion of more AT devices to compensate for specific functional needs.

*I think the app could have more information. . . The equipment seems like things that one is practically tired of seeing the same. I didn't see new equipment that I know exists, better things. I could give you an example of what struck me the most due to my back condition that prevents me from using a mop. What I saw was a mop-type vacuum cleaner but with the same movement as a mop, which is not feasible for me to use. Currently, there are robots that, in my opinion, should have been included. Modern things in terms of new equipment that exists. Update in terms of equipment. (L1)*

Quality of visual information (uMARS item 15; M = 4.33, SD = 0.65). This sub-theme refers to whether the photos and videos of the MGAT are clear, logical and correct. Most of the participants agreed with the following: "Everything is understood very well. It is summarized and they provide a video for you to see it in action" (B2). This consensus was confirmed by a moderately high uMARS item 15 mean score. However, the lack of audio descriptions in the videos was mentioned again as a factor affecting the videos' quality.

Trustworthiness (uMARS item 16; M = 4.67, SD = 0.65). This sub-theme pertains to the belief that the information provided in the MGAT comes from a reliable source, confirmed by a high uMARS item 15 mean score. All participants agreed that the MGAT's information is reliable, primarily because it was developed and tested by a trustworthy academic institution: "That's from the Medical Sciences Campus, so that information is really good" (B1).

### 3.3.6. Useful

The "useful" theme exemplifies participants' perception of the MGAT's ability to fulfill older people's functional health needs, encompassing the two sub-themes described below.

Useful to Others (uMARS item 17; M = 4.25, SD = 0.97). All participants stated that they would recommend the MGAT to others because they thought that other older adults could derive benefits from this web app: "I would definitely recommend it to everyone because it is an essential help for many. . . a splendid help for dealing with things, and Puerto Rico is aging at a rate that surprises people" (N3). Others would only recommend the MGAT to people who they think might find its use useful: "If it's for people who would benefit from it, I would definitely recommend it. . . I have already done it for several people who need it. . . telling them about this application that exists and that can help them" (L2). This was confirmed by a moderately high uMARS item 17 mean score.

Useful to self (uMARS item 18; M = 3.58, SD = 1.00). Most participants agreed that they would continue using the application for the next 12 months because they could derive benefit from its use, which was confirmed by a moderately high uMARS item 18 mean score. This was expanded by the participants' reasons for continuing to use the MGAT. Self-management of their functional needs was the most reported motivation for its use: "Because it helps to manage health conditions. . .the limitations that come with age" (N1). To support their lifelong learning and memory of the AT devices was also a common motivation: "I will continue using the application always. . . it motivates me to have the knowledge of devices that can help with age" (A2) and "I would use it 3 to 10 times (in the next year) to be able to look from time to time and refresh my memory of things I can do" (N2). Some participants also had the intention to buy or even acquired various AT devices to self-manage their functional limitations: "The reacher caught my attention, and I bought it because I liked it a lot. I'm about to buy the nail clippers" (N1).

### 3.3.7. Valuable

The valuable theme comprises the delivered value of the MGAT to the participants, with two sub-themes emerging: "worth paying" and "empowered independence".

Worth paying (uMARS item 19; M = 2.50, SD = 1.93). There were mixed opinions about the participants' willingness to pay for the MGAT, confirmed by a moderately low uMARS item 19 mean score with a high SD. Several participants indicated that they would pay for the MGAT given its value to support their functional health: "Definitely, I would pay to get the app because it is helpful for functioning, for people like us from the elderly community" (N1). In contrast, most of the participants stated that they would not pay for the MGAT because they could obtain information about AT devices through other sources: "I would definitely not pay because I understand that with a couple of times that one has looked at it, you already more or less know the needs that a person with disabilities might have and I go to Google, that gives me the information and helps me to have the device". Moreover, four participants reported that they preferred the application to be free of cost: "As long as I can get it for free, better" (B3).

Empowered independence (uMARS section F Perceived Impact; M = 4.70, SD = 0.81). Most of the participants valued the MGAT's usefulness to increase their awareness and knowledge of AT devices that could help them, as well as other people, maintain their autonomy to independently perform, self-direct, and participate in meaningful daily activities. The participants talked about the value of the MGAT to support their autonomy and safety at the present time: "It gives me safety and helps me continue doing things by myself" (B1). The participants also reflected on how the MGAT was valuable to proactively know about AT devices that they could need in the future when experiencing functional decline:

> *This app has an immeasurable worth because it teaches you many things and devices that you were unaware of, and it gives you knowledge on how to help yourself in the future. Even though you might not need some of the devices at the moment, if you are on the path to needing them, you can, according to the application, select which device will be beneficial according to your problem. (A2)*

Those with higher functional abilities valued the knowledge gained through the MGAT that allowed for them to inform significant others about AT devices that could help them overcome their functional needs: "It gives me information, and I can know more about the devices, and I can also help my mom who needs them" (B3). Finally, two participants talked about the particular value of the MGAT to maintain autonomy and dignity among those who live alone.

> *For me (the MGAT) is worth a lot. Because many people who live alone don't have someone to remind them, 'Hey, Dad, did you take your pills?' 'Look, Dad, don't walk like that, you're going to fall'…because you see it as scolding. In the application, you see it as someone who is giving you advice. It's not scolding…the application speaks to us with respect…that is a splendid for moving on. And that loneliness of feeling 'I am alone', there (MGAT) you have a companion to help with 'how to go to the bathroom… how to go down the stairs. (N3)*

The participants' voices regarding empowered independence were confirmed by a high uMARS Section F Perceived Impact mean score.

## 4. Discussion

This study investigated the quality of the MGAT web app for older Latinos with physical function disabilities by its users and explored the experiences of older adults using this AT web app. Participants considered the MGAT to be of good quality, with a mean uMARS quality score of 3.99 on a 5-point Likert scale. The functionality, aesthetics, information content, and perceived impact of the MGAT were particularly well appreciated, with uMARS scores above 4. The subjective quality of the app was also recognized (uMARS score 3.6). Qualitative data largely showed a positive reaction to the use of such a web app,

with numerous perceived benefits reported. The MGAT was mostly found to be findable, accessible, usable, useful, credible, and valuable. However, further improvements to the MGAT's desirability are required.

Other AT apps have been studied to support the self-management of functional disabilities among older people, demonstrating benefits in areas such as prospective memory functioning, independence, autonomy, spatial orientation in unfamiliar environments, and self-management and cognitive rehabilitation in older adults with mild dementia [39,40]. Some studies have also explored the feasibility and acceptability of apps for managing disabilities following a stroke or head injury [41,42]. However, these apps have certain limitations that the MGAT seeks to address. Firstly, many existing apps are disease-specific, targeting individuals with conditions like dementia or stroke, which limits their applicability to a broader population of older adults with various functional needs. The MGAT, in contrast, provides information for a general population of older adults with non-specific physical function limitations in activities of daily living (ADLs) and instrumental activities of daily living (IADLs). Secondly, these apps are often developed for North American, European, and Asian populations, lacking representation of Latino users. The MGAT stands out by offering culturally relevant content and videos featuring older adults similar to the target Latino app users. This cultural relevance is crucial, as culturally tailored health interventions are considered a primary approach to addressing ethnic health inequalities [43] and improving adherence to treatment plans [44].

Our participants highly valued the usefulness of MGAT in increasing their awareness and knowledge of AT devices and services that could help them, as well as other people, maintain their function and autonomy in performing daily activities. This aligns with the active aging framework of the World Health Organization, which acknowledges that lifelong learning is a crucial factor facilitating participation, health, and security as people grow older [45]. Moreover, increasing older adults' knowledge of AT is vital, as research has documented that information about how to acquire AT devices is the most important prerequisite for the use of AT [8]. Although the actual functional benefit from engaging with MGAT can only be determined through a clinical trial with the web app, MGAT shows potential to decrease existing disparities in the prevalence of functional disabilities in two ways. First, by addressing the most prevalent barrier for using AT among older Latinos—the lack of access to information and knowledge of AT [17,18]. Second, by motivating older Latinos to acquire and use AT devices, as seen by some participants in this study, given its perceived benefit for improving their function.

Participants' feedback has been instrumental in improving the MGAT. When developing instructional videos for older adults, providing adequate instructions is important to ensure users can complete a desired task, particularly when the task is novel [46]. Although the quality of the videos was highly rated by some participants in our study, the absence of audio instructions in the MGAT videos had a negative impact on the perceived desirability and credibility of the MGAT for some participants. The lack of multimedia video instruction specifically affected its appropriateness for older adults, the quality of the videos, and its potential to be entertaining. This is consistent with recommendations in the existing literature stating that older adults obtain greater benefits from videos with audio instructions rather than using a single instructional modality [46,47].

Consistent with the findings of a previous study on the evaluation of health apps by adult users [48], some participants tended to reduce their usage of the MGAT when no new information was offered. To sustain app engagement, it is recommended to provide regular notifications of new AT devices. This will not only enhance interactivity with the MGAT but also assist individuals with memory issues in reviewing information about AT devices on the MGAT.

In terms of the methodology used in this study, there is a limited number of studies that have assessed the quality of a system using a mixed-methods approach over an extended period, as recommended in the current literature [49]. Our study takes a comprehensive approach to examining the experiences, needs, and preferences of AT apps

among community-dwelling older adults with physical function disabilities. By employing a mixed-methods approach, our research team was able to pinpoint crucial aspects of health app features and functionalities that could have a positive impact on the future development of AT apps.

This study has a few limitations that must be considered when interpreting the results. Firstly, the descriptive study featured a sample size that is sufficient for a basic usability assessment but is not large enough to draw definitive conclusions about the perceived quality and experience of AT apps to inform about AT devices and services supporting the function and independence of older adults with physical function limitations. Additionally, as the participants had an average mean level of physical function limitations and were Latino older adults, the results cannot be generalized to other older adult population groups, particularly those with lower levels of physical function and from different ethnic and racial backgrounds.

Regarding future work, the MGAT is undergoing further development to incorporate participant recommendations, such as improving instructional videos, adding notifications of new AT devices and links to acquire them, and increasing its degree of personalization. Future studies should assess the MGAT's quality and experiences from different populations to achieve a more general perspective (e.g., involving older adults, informal caregivers, or healthcare professionals). Moreover, future work should conduct pragmatic trials to test the effectiveness of the MGAT on increasing older Latino function in daily activities by enhancing their awareness, knowledge, motivation, and use of AT devices and services.

## 5. Conclusions

Overall, users highly rated the quality of the MGAT, expressing a positive experience with the web app. While some features need improvement, this application has the potential to increase the knowledge and adoption of AT devices among the vulnerable population of older Latinos with high levels of functional disabilities but limited access to AT interventions.

**Author Contributions:** Conceptualization, E.M.O.-C. and R.M.-P.; methodology, E.M.O.-C., A.F.-T., N.F.-A., B.O.-V., N.L.R.-R., G.A.T.-F. and R.M.-P.; formal analysis, E.M.O.-C., A.F.-T., N.F.-A., B.O.-V., N.L.R.-R. and G.A.T.-F.; investigation, E.M.O.-C., A.F.-T., N.F.-A., B.O.-V., N.L.R.-R. and G.A.T.-F.; data curation, E.M.O.-C.; writing—original draft preparation, E.M.O.-C., A.F.-T., N.F.-A., B.O.-V., N.L.R.-R. and G.A.T.-F.; writing—review and editing, E.M.O.-C.; supervision, E.M.O.-C.; project administration, E.M.O.-C.; funding acquisition, E.M.O.-C. All authors have read and agreed to the published version of the manuscript.

**Funding:** This research was supported by Hispanics-In-Research Capability (HiREC)—National Institutes of Minority Health and Health Disparities (NIMHD) under Award Number S21MD001830, and The Hispanic Alliance for Clinical and Translational Research (Alliance) supported by the National Institute of General Medical Sciences (NIGMS) National Institutes of Health under Award Number U54GM133807.

**Institutional Review Board Statement:** This study was conducted in accordance with the Declaration of Helsinki and approved by the Institutional Review Board of the University of Puerto Rico Medical Sciences Campus (protocol number 2211060074, approved 19 December 2022).

**Informed Consent Statement:** Informed consent was obtained from all subjects involved in this study.

**Data Availability Statement:** Due to restrictions, the data presented in this study are only available upon request from the corresponding author. The data are not publicly available.

**Acknowledgments:** We would like to express our sincere gratitude to Wovenware for their invaluable contribution to the design and development of the My Assistive Technology Guide (MATG) web app.

**Conflicts of Interest:** The authors declare no conflicts of interest. The funders had no role in the design of this study; in the collection, analyses, or interpretation of data; in the writing of the manuscript; or in the decision to publish the results.

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
