# Peer review of "Empowering Potential of the My Assistive Technology Guide: Exploring Experiences and User Perspectives"

_disabilities, doi:10.3390/disabilities4020020_

Round 1
Reviewer 1 Report
Comments and Suggestions for Authors
An interesting paper on a relevant and ongoing topic of importance.
A few minor things:
The title says Latino but the majority of the rest of the paper uses Hispanics. Would be good to be consistent, or state that the two are used interchangeably.
Would be useful to mention Puerto Rico in the abstract
line 31 - 'unctional' needs removing?
Interested to know why 30 days was chosen for the user trial. Was any note made of how often it was used? Is it imagined to be something people will use often?
How is it proposed that the app will be made available to older people? How will they be encouraged to use it? (this may be beyond the scope of the current stage of the project)
Is there going to be ongoing support for the app to ensure it is always up-to-date (in terms of prices, availability, etc)?
What is the difference between 'high school graduate' and 'something higher than high school' (something higher suggesting they graduated on from it)
What was the procedure for translation of interviews to English? How were translations checked and verified for accuracy etc for the qualitative analysis?
Is there scope to include some images from the app within the paper?
Reviewer 2 Report
Comments and Suggestions for Authors
Thank you for the opportunity to review this interesting and well-prepared manuscript.
These authors have developed an app to assist older people learn about assistive products they might benefit from. This work explores the usability of that prototype using a battery of tests from ICT and user-interface literature. The method is clear, robust, and the findings are sound. They report that the app is promising for older people.
I am hopeful this paper can be published, and am sure it will be a welcome contribution. I have recommended a minor review.
As I describe below, my main concern is strengthening the description of the app itself, and situating it in the literature concerning user-led decision-making about AT need, and of population AT needs. Overall, we learn that the app is promising and helpful for older people, but less about what it actually does. Do people make different consumer choices, seek more information? Are there any concerns people will make the wrong choices. Or, have the authors limited the list to relatively safe and general products, avoiding products like wheelchairs, mobility devices, some canes, vision products and others, that would ordinarily require some clinical intervention.
Below, I offer more detailed comments. Thank you to the authors for their important work.
Title.
I think 'transcending function' is a bit aspirational, and doesn't add very much to how the paper will be found in relevant searches. The title should simply name and describe the study. For example, Empowering Potential of the My Assistive Technology Guide: Exploring experiences and user perspectives.
Abstract.
Specify that the awareness is mostly potential AT users.
Keywords: is functional disabilities the right keyword?
31. Typo unctional.
31. I find the term 'functional disabilities' unusual and unexpected, and certainly the acronym AD is not one I have seen before. The definition given seems to describe 'functional difficulties'. A disability is defined by some interaction with the environment. This might be a case of a preferred usage among Hispanaphone authors - and I do defer to the authors here. However, I invite them to consider whether functional difficulties or just 'function' as relevant in context, is a better term.
35. Functional difficulties are also associated with injury.
55. The rationale of this paper and the intervention it describes is premised on knowledge about AT being low, and a key driver of non-use of AT. A fairly old paper is cited, where recent epidemiological and quantitative work very much supports and strengthens this assertion. I suggest the authors find recent quantitative work of assistive product unmet needs from population surveys, especially those that highlight knowledge about AT being low. Further, knowledge is only one driver of under-use of AT. it is described as the 'predominant driver among hispanics', whereas the paper cited does not concern hispanics, and knowledge is only one determinant of unmet AT use (albeit a very important one).
150. I don't think this image adds very much to the excellent text description of the parallel method.
188. Each meeting was facilitated by five students? Or, meetings were facilitated by one of the five?
I think somewhere about here, the intervention itself (the app) needs to be described more completely. The description in the background/introduction could be shortened a bit, with a fuller description in the methods. Or, if the authors feel strongly for some reason that the intervention is not part of the work per se, the description in the background needs to be a bit more complete. In short, it's hard to know what the app actually does beyond the general descriptions provided.
295. this is really a question for the editors, but I think this table might span several pages unnecessarily. The information is important so I think if it were divided into women and men, it might fit more effectively while having a local split of groups. Or, consider whether table 3 could add some more columns and have a mean/median row at the bottom, and remove table 2 entirely. I think with this number of cases, it might make sense to have a case-by-case table. Any information that is less important and might not fit in the table could be added to the running text.
321. Is a mean the right statistic here? Wouldnt a frequency distribution make more sense? I do think the mean and range give enough information but do consider whether this is the right way to report uMARS (and perhaps cite references if relevant).
459. My main question about this paper is how decisions are made about which person might need a product. The authors will understand that many products might be desirable to a consumer, but not always appropriate. (it is equally plausible that clinically recommended products don't suit consumers of course). More information is required about what information users enter, and the basis on which devices are chosen. How, if at all, are users instructed to proceed? Are there situations where the user is guided to a clinician for support with that decision-making? Has any content validation been completed?
There are two reasons for this. First, readers will expect clarity on how those choices are made. Second, a user-guided approach vs a clinically-led approach is a contested area.
Consider the WHO rATA as an example of a simple user-informed estimate of AT need, compared with clinical screening. The former is simple, user-led, while the latter is more complex and requires some clinical expertise and technology. The two produce very different estimates of need. Put simply - is the app making the right guidance? How do you know, and how does the reader? I think this needs to be explored in the discussion as well. This relates to credibility and trustworthiness dimensions in the results.
Overall I find the results section very long. To the extent possible, shorten it.
594. I think it would be interesting to explore apps that use simple algorithms and decision-trees to determine needs for disability/functioning related needs along with others. There is some emerging work in this space, including community based case management and screening (of a range of needs, including assistive technology).
662. The conclusion is long and repeats information from the discussion. Move future work to discussion.
667. This is the first use of 'high functional needs' in the paper. Consider.
I think the link to the prototype makes much more sense in the methods (see my comments about describing the app and how it works). It probably also makes sense as 'supplementary material'.
Round 2
Reviewer 1 Report
Comments and Suggestions for Authors
Thank you for your comprehensive consideration of the comments made.
Reviewer 2 Report
Comments and Suggestions for Authors
Thank you to the authors for their careful responses to the reviews.
I am satisfied the authors' have addressed my concerns. They have gone further still, and made some interesting and valid improvements to the paper. This has improved an already strong manuscript.
I am recommending the manuscript is now accepted for publication provisional on the usual editorial revisions.